# Microwave Imaging of Breast Skin Utilizing Elliptical UWB Antenna and Reverse Problems Algorithm

**DOI:** 10.3390/mi12060647

**Published:** 2021-05-31

**Authors:** Sameer Alani, Zahriladha Zakaria, Tale Saeidi, Asmala Ahmad, Muhammad Ali Imran, Qammer H. Abbasi

**Affiliations:** 1Center for Advanced Computing Technolgy (C-ACT), Faculty of Information and Communication Technology, Universiti Teknikal Malaysia Melaka, Hang Tuah Jaya, Durian Tunggal, Melaka 76100, Malaysia; asmala@utem.edu.my; 2Faculty of Electronic and Computer Engineering, Universiti Teknikal Malaysia Melaka (UTeM), Durian Tunggal, Melaka 76100, Malaysia; 3Electrical and Electronic Engineering Department of Universiti Teknologi PETRONAS, Bandar Seri Iskandar 32610, Perak, Malaysia; tale_g03470@utp.edu.my; 4James Watt School of Engineering, University of Glasgow, Glasgow G12 8QQ, UK; muhammad.imran@glasgow.ac.uk (M.A.I.); Qammer.Abbasi@glasgow.ac.uk (Q.H.A.); 5Artificial Intelligence Research Center (AIRC), Ajman University, Ajman 346, United Arab Emirates

**Keywords:** UWB antennas, skin cancer, reverse problems, microwave imaging

## Abstract

Skin cancer is one of the most widespread and fast growing of all kinds of cancer since it affects the human body easily due to exposure to the Sun’s rays. Microwave imaging has shown better outcomes with higher resolution, faster processing time, mobility, and less cutter and artifact effects. A miniaturized elliptical ultra-wideband (UWB) antenna and its semi-spherical array arrangement were used for signal transmission and reception from the defected locations in the breast skin. Several conditions such as various arrays of three, six, and nine antenna elements, smaller tumor, multi-tumors, and skin on a larger breast sample of 30 cm were considered. To assess the ability of the system, a breast shape container with a diameter of 130 mm and height of 60 mm was 3D printed and then filled with fabricated skin and breast fat to perform the experimental investigation. An improved modified time-reversal algorithm (IMTR) was used to recreate 2D images of tumors with the smallest radius of 1.75 mm in any location within the breast skin. The reconstructed images using both simulated and experimental data verified that the system can be a reliable imaging system for skin cancer diagnosis having a high structural similarity index and resolution.

## 1. Introduction

One of the major causes of death can be assumed as skin cancer. Numerous types of skin cancer exist and are known as basal cell and squamous cell carcinoma, and melanoma. Among these three, melanoma is known as the deadliest. It should be detected before it becomes unpredictable to prevent the fast growth of cancerous cells. It can be detected as changes in dimensions, color, and texture of the examined skin [1].

Numerous imaging methods have been used for skin cancer imaging. These techniques are known as ultrasound (US), which can detect a tumor in vivo with a resolution of 80 mm and 200 mm for lateral and axial resolutions, respectively [2]. However, it cannot show the differences between healthy and unhealthy skin [3]. Another method known as magnetic resonance cannot produce suitable and reliable information about tumors with more than a 15 mm diameter. Furthermore, the device is a costly, complex, time-consuming process, and is not comfortable for patients.

The near-infrared, known as the NIR method, depends on backscattered photons to construct the image of the tumor. However, it has some limitations in decent signal penetration into the skin. Another applied technique for the imaging of skin is confocal microscopy (CM), however, this suffers a lot due to the limited penetration in the skin [4,5,6].

Microwave imaging (MI) offers reconstructed images by using the alterations in dielectric properties of the sample under test (SUT) (skin). For instance, a contrast from 2 to 10 times exists between the healthy and unhealthy skin, about 2 to 10 times higher [7]. Thus, there will be high reflection when the signal hits the tumor. Microwave imaging can benefit us due to its excellent characteristics and advantages such as low stake of health, sensitivity to dielectric changes, early detection of tumors, non-invasive, and simple to perform, high resolution images [8,9,10,11].

Several works have been carried out to detect tumorous cells in the human body utilizing microwave imaging techniques. Breast and skin cancer have shown quite an interest to be investigated by researchers among all the medical purposes. For instance, an antipodal Vivaldi ultra-wideband (UWB) microwave system for breast cancer detection is presented. The proposed system included a monocycle pulse generator, antipodal Vivaldi antenna, breast model, and calibration algorithm for tumor detection [12]. A Flexible Broadband Antenna and Transmission Line Network designed at a frequency range of 2–4 GHz for a Wearable Microwave Breast Cancer Detection System [13].

In addition, skin cancer has been of significant research interest since the number of people affected by skin cancer has increased dramatically. Both microwave and millimeter-wave frequency ranges have been used to detect tumors in the skin. An ultra-high-resolution imaging system was used to detect tumors in the skin in the mm-wave region. They used a Vivaldi tapered slot antenna with SIW and the coplanar feeding technique. Despite detecting a tiny tumor having high resolution, the antenna system was complex, and the RF components were expensive at those high frequencies compared to the microwave region RF components [14]. A flexible AMC antenna designed for early detection of skin cancer was designed at the X-band and frequency range of 8–12 GHz, the radiation efficiency of 65%, and maximum gain of 6.5 dBi (limited bandwidth so a low resolution was obtained) [15]. Another AMC wearable UWB antenna was designed at the X-band from 8 GHz to 13 GHz to detect a tumor in the skin. It presented a maximum gain of 3.2 dB and dimensions of 44 × 48 mm^2^ (limited bandwidth so a low resolution was obtained) [16,17].

An antenna used the concepts of metamaterial (MTM) and substrate integrated waveguides (SIW) at the terahertz (THz) frequency band. It was designed on a transparent substrate of polyimide and aluminum as a conductor and obtained the BW of 0.6_0.622 THz with dimensions of 1 × 1 mm^2^ [18]. Arrays of a wideband antenna designed with low sidelobe, high gain of 16.8 dB, and BW of 8.1–12.8 GHz [19]. The MTM structure was used to improve the radiation characteristics of the antenna with a small size of 38.5 × 36.6 mm^2^ and attained the BW of 0.55–3.85 GHz and gain of 5.5 dBi [20]. In addition to these studies, a UWB antenna used the concept of the MTM structure to improve the gain and efficiency by 4.8 dBi and 18%, respectively. Furthermore, it offered a BW of 10 GHz [21].

Before the tumor’s image reconstruction started, some antenna elements were required to be positioned for both arrangements of planar and semi-spherical. A semi-spherical array elements arrangement was utilized throughout the investigation of the breast skin.

This paper is configured in five parts. First, the skin cancer’s background, the challenges, and the applied techniques are presented. Then, the microwave technique was introduced as an alternative. Then, an algorithm is recommended to improve the image reconstruction accuracy and clutter removal. Then, the simulated and measured data were used to reconstruct the image. The paper is concluded in the last section.

## 2. Microwave Imaging Modeling of Breast Skin

Microwave imaging is a promising method in medical imaging consisting of three kinds of passive (working based on heating and temperature), hybrid (a combination of microwave and acoustics), and active (electrical properties of SUT) [7]. A UWB antenna, as an important part of every imaging system, sends and receives a UWB pulse to create an image of the target [22,23,24,25,26,27,28]. After obtaining the received signals, an algorithm is required to reconstruct the image of the tumor. This algorithm should remove the clutter and artifact effects in order not to mask the tumor response and emphasize it for image reconstruction. Several algorithms such as data-adaptive and data-independent were used to reconstruct the image of the tumor in the breast skin. Since both groups showed some drawbacks, a mixture of both is proposed to improve them to have a high-resolution image [29,30].

### 2.1. Arrangements of Antenna Array Elements

The excitation, simulation, and measurement were performed when the UWB antennas were in contact with the breast skin. A semi-spherical breast shape container made of polylactic acid (PLA) was 3D printed to hold the set of antenna array elements in the required locations. Figure 1 shows the simulated and modeled proposed antenna array elements along with the breast skin and the PLA semi-spherical plastic container. The container was used to maintain the antenna elements oriented toward the skin breast; this was completely fixed and does not move during the experiment. The antennas illuminate the skin once they are connected to the cables and signal generator. It should be mentioned that a space of 5 mm was considered between each array of elements and the breast skin.

The system’s structure can adapt to breast skin with a minimum and a maximum diameter of 100 mm ($r_{min}$) and 300 mm ($r_{max}$), respectively. Every two antennas were separated with a radial distance of 100 mm (300 mm for the larger sample), 20 mm vertically, and 40 degrees horizontally. Therefore, if A is the antenna numbers from 1 to 9, the position of other arrays can also be positioned accordingly in polar coordinates (Equation (1)).
(1)$$\gamma =r_{max}-d,\;\theta =\left(A-1\right)2\pi /9$$
where d is the radial displacement of arrays in terms of the maximum radius.

The plastic tape was made of PLA to reduce the impacts on the electromagnetic (EM) field distribution. The PLA has a dielectric constant close to air; thus, it affects the EM fields minimally. To illuminate the breast skin and collect the scatter electric fields and signals, a set of nine monopole elliptical UWB antennas with dimensions of 15 × 15 mm^2^ was utilized (Figure 1b) (when the antenna dimensions are also small, multiple antenna elements are permitted to gather more data from scattered signals). Furthermore, the microwave radar technique allows for the reduction of clutter from highly scattered signals from targets (tumors) for image reconstruction, [23]. The antenna was proposed and showed promising performance in the breast skin. The antenna was designed on a substrate with a dielectric constant of 2.55. The UWB antenna used for this investigation had an operating bandwidth of the proposed UWB antenna, which obtained a fractional bandwidth of almost 26.1 GHz [31]. Thus, it will be a good candidate for imaging purposes since it has broad bandwidth (wider bandwidth gives a higher range resolution).

Some parameters should be discussed before starting the modeling section of the antenna’s modeling and the imaging ability. These parameters are isolation among antenna array elements, transmission coefficient, reflection coefficients, working BW, gain, and radiation efficiency. The isolation among the array elements can be shown by the transmission coefficient of the array elements and shows how the array elements affect and interact with each other. The Sn1 factor shown in [31] showed that the array elements were in acceptable isolation since the Sn1 was less than −18 dB at most of the working BW of 3.9–30 GHz. Furthermore, the antenna obtained a peak gain of 6.48 dBi and radiation efficiency of more than 90%. All these parameters have already been published and explained by the authors in [31]. Therefore, they are not explained here, and only the electromagnetic modeling and behavior are presented.

### 2.2. Modelling of Electromagnetic Behavior

When a time reversal (TR) algorithm is used, the communal characteristics of the wave equations come to mind. This means that both electric and magnetic fields are backpropagated to concentrate on the source and the target (transmitters and receivers are like mirror). This algorithm has been applied for numerous applications using ultrasound imaging [32] and showed promising outcomes, especially in heterogeneous, cluttered, high-dense environments [33,34,35,36]. The TR overtook the other algorithms, which were time-consuming, sensitive toward noise and interference, and low resolution [37]. However, it encountered several challenges during image reconstruction such as undesired backscattered signals from the target, is complicated, and it needs to know the location of the antenna elements. In the communication and/or imaging system, the correlation between the transmitted and received signals should be high not to lose important data and information [38].

The electromagnetic waves in the medium under test should be studied before using the TR algorithm. As a result, the Maxwell equations are used. The propagation of electromagnetic energy away from time-varying sources (current and charge) in the form of waves is predicted by Maxwell’s equations (Equation (2)). In a source-free field, consider a linear, relatively homogenous, isotropic medium categorized by (*μ*, *ε*, *σ*), which can be described as
(2)$$\nabla \times E=-\mu \frac{\partial H}{\partial t},\;\nabla \times H=\sigma E+ℇ\frac{\partial E}{\partial t},\;\nabla .E=0,\;\nabla .H=0$$

By studying the solutions to the wave equations that describe the electric and magnetic fields of the wave, the characteristics of an electromagnetic field can be calculated. The wave equation can be solved and described using Maxwell’s equation in a lossless and synchronous medium as:(3)$$\left(\nabla^{2}-\mu \varepsilon \frac{\partial^{2}}{\partial t^{2}}\right)A\left(r,t\right)=0$$
where *A* (*r*, *t*) and *A* (*r*, *t*) in Equation (3) gives two diverging and converging solutions (*r*, −*t*). As a result, to return the resource, the diverged wave from the point source is backpropagated using its original direction. *A* (*r*, −*t*) is a time-reversed field that provides a spatial concentration when backpropagated in configurable media. The IMTR is then applied to measure and reverse targeted fields, after which the next two simulation steps are numerically determined. CST Studio 2019 performs simulation studies and models using the 2D finite integration technique (FIT). The modeled UWB antenna sensors can send and receive signals. During the simulation, the boundaries were treated as perfectly matched layers, and their boundary equations were derived using Maxwell’s equations (Equation (2)). The time-reversed fields are back-propagated after the FIT equations have been solved.

The antenna elements input signal was assumed as a regulated ultra-wideband Gaussian pulse having an operating frequency of *f_r_* = 2.4 GHz, defined as Equation (4).
(4)$$S\left(t\right)=e^{-{\left(t-t_{0}\right)}^{2}/2\omega^{2}}cos\left(2\pi f_{r}t\right)$$
where *t*_0_ and ω are the center and width (78.1 ps) of the pulse, which is related to a half-power bandwidth of 25 GHz, as depicted in Figure 2.

The total field and signal of the received signal should be considered for both with and without tumor before the simulation begins (Figure 3). The antenna array elements were positioned around the breast skin with a radius of 50 mm to attain each received signal between every two array antenna elements. A central spherical tumor with a radius of 1.75 mm was assumed within the semi-spherical breast skin sample. The transmitter transmits the Gaussian pulse in Equation (4) and Figure 2 and is then received by the other array elements.

## 3. The Proposed Algorithm for Image Reconstruction

Planar array arrangements have been applied to detect tumors in the breast [39,40]. The proposed study used a semi-spherical imaging environment with a diameter of 100 mm and height of 60 mm and the tumor considered in the skin (1.75 mm radius) located on the breast media.

The imaging reconstruction starts with a calibration process for the received signals. The signal reversed in time for both with and without tumor in the breast. The reversing in signals is to focus on the tumors and emphasize their response. It also helps to reduce the clutter and artifact removal, since the tumor response is affected and dominated by other interference like the clutters in the imaging environment. Figure 4 shows the proposed imaging algorithm.

### 3.1. Calibration of the Signals

In other parts of the imaging environment like other sources, sidelobe coupling impacts other than scattering signals affect the tumor response. To perform the calibration, both background signals (*E*_1_ signals when no breast phantom exists) and scattered signals (*E*_2_) are considered, then (*E*_2_) is subtracted from *E*_1_ to achieve the calibrated *E*_3_.
(5)$$E_{3}=E_{2}-E_{1}$$

The output from Equation (3) yields reduced clutter. When the background signal is deducted and creates the calibrated output, it is multiplied by a weighting factor. The calibrating signals are emphasized by weighting factor multiplication, and their averaged signals are deducted from *E*_3_ to obtain the early content removal output (*E*_4_). Signal averaging provides early content removal. Furthermore, utilizing signal averaging and time gating can provide smooth signals [41].

### 3.2. Paired Multiplying Scattered Output

The output *E*_4_ is then paired multiplied for each scattered signal from each antenna array element to improve the accuracy [42]. If the number antenna elements are shown by n and each signal is represented by S, the paring multiplication of these signal can be as follows:n = 1, …, 9.(6)
y1 = S1.S2, y2 = S1.S3, y3 = S1.S4, y4 = S2.S3, y5 = S2.S4, y6 = S3.S4, ….(7)

Therefore,
(8)$$E_{2}{}_{s,m}\left(t\right)=\left[y1\;y2\;y3\;y4\;y5\;y6\;\ldots \right]$$

When the scattered signals are multiplied, they are then averaged once more to suppress the undesired clutters and enhance the chance of tumor detection (*E*_5_).

### 3.3. Filtering the Output

To filter the output obtained from the last section (*E*_5_), the time of arrival was defined for carrying out the time grating and localizing the tumor. Additionally, it improved the clutter suppression further and unmasked those areas of the image that were still smeared due to the ineffective clutter and artifact removal. It measured the time when a wave reached a breast skin’s front and back walls and are known as early-time ($t_{E}$) and late-time ($t_{L}$), respectively. It provides a suitable window that can decrease the clutter more and it can be defined as follows:(9)$$E_{6}=\left\{\begin{array}{cc} & \;\;\;E_{5}\;if\;t_{E}<t<t_{L}\\ 0, & otherwise \end{array}\right.$$

Figure 5 shows the procedure used for gating and time of arrival calculations.

The achieved *E*_6_ from the previous step is then multiplied by Gaussian pulse to make the detection of even tinier tumors possible. Equation (5) is to calculate this Gaussian pulse.
(10)$$E_{7}=E_{6\cdot }\cdot e{\;}^{-{\left(\left(t-tp\right)\;/\;\tau \right)}^{2}}$$
where *tp* is the apex and τ is the tapering factor in the Gaussian window. *E*_7_ is the output obtained known as the proposed MITR.

## 4. Both Simulation and Measurement Data Used for Image Reconstruction

The data were imported to reconstruct the image after the important and effective factors in imaging such as received signals, fidelity, and the coupling and isolation effect between every two arrays obtained. The significance of the proposed algorithm and imaging system was investigated to detect a tumor in breast skin using both simulated and measured data using four separate trials. These trials were considered as the number of antenna elements, smaller tumor (2.5 mm diameter), two and three tumors in the imaging media, and larger breast skin phantom with a diameter of 30 cm. A tumor with a radius of 2 mm was assumed in all imaging trials except for the one with a 1.75 mm radius. Besides, the radius of the breast was 50 mm for all investigations except for the one with a radius of 15 cm.

The proposed algorithm was used to reconstruct the image of the tumor after importing all the received signals and calculating the delay time between every two-array element. Then, the antenna element’s positions and their distances from each focal point were introduced. The intensity value of each point was defined, and then the image was formed and reconstructed. It was comparatively evaluated to depict the improved modified algorithm’s capability to reconstruct an image against three conventional algorithms: modified weight delay and sum (MWDAS), faster delay multiply and sum (FDMAS), and standard time reversal (STR).

### 4.1. Reconstructed Images Using Simulation Data

The first investigation to check the imaging system’s capabilities in the detection of tumors was to investigate the effects of array numbers around the sample. Figure 6 shows the constructed image using three arrays of the antenna sensor around the sample using MWDAS, FDMAS, TR, and the proposed algorithm. The improved algorithm showed the best outcomes compared to the other algorithm. MWDAS showed too much clutter and artifact around the actual sample, the image was not clear, and the accuracy was too low. When the algorithm changed to the FDMAS algorithm, the accuracy and the clutter removal was improved. However, it was not too clear, and the detected image was faded. Therefore, the TR algorithm was used due to its capabilities in the detection of scatterers in a dense and cluttered media like skin and the breast fat next to it, but it still detected some clutter around the tumor. Thus, the improved algorithm was utilized, and it showed the best results compared with the others, as shown in Figure 6.

Figure 7 demonstrates the constructed image using six arrays of the proposed antenna sensor around the sample to detect the tumor. The same trend was followed in Figure 7 as the improved algorithm had the best results, among others. Besides, the constructed image using six arrays around the tumor showed better quality and accuracy in localization and clutter removal compared to the obtained results using three arrays around the tumor shown in Figure 6.

After obtaining the results using three and six arrays, the array numbers were increased to nine array elements to achieve more accuracy and remove more clutters and artifacts around the tumor. Figure 8 illustrates the constructed image using nine array elements around the tumor using the MWDAS, FDMAS, TR, and the improved algorithm. Like the results obtained from the other array numbers, the same tendency was followed when the arrays were enhanced to nine array elements. As is shown in Figure 8, when the arrays were nine, the results were the best compared to the others. The improved algorithm showed the best outcomes in detection and localizations in comparison with the others.

The next investigation to check if the proposed system could detect the tumor in the skin located next to the breast fat was to evaluate whether it could detect the tumor when the tumor was smaller with a diameter of 5 mm. Figure 9 shows the reconstructed image with nine arrays around the breast and tumor when the tumor diameter was 3 mm. Furthermore, the improved algorithm showed the best results, and it could detect the tumor. Using the improved algorithm gave us the exact location, and the artifact was perfectly removed. The TR algorithm could detect the tumor but not at the actual location of the tumor. Both MWDAS and FDMAS detected a faded tumor and the location of the tumor was not accurate.

The system capabilities should be checked when more than one tumor exists. Thus, we evaluated when two and three tumors existed around the sample. Figure 10 shows the reconstructed image when two tumors exist around the sample. Same as that shown in other investigations, the improved algorithms showed good accuracy in the detection of both tumors. Both MWDAS and FDMAS could not detect the tumor very well. The position of both tumors was not the actual tumor in the simulations. The TR algorithm showed better results than MWDAS and FDMAS. However, more clutters and artifacts existed compared to the improved ones.

After showing that the system could detect two tumors in the skin in the presence of breast fat, one more tumor was added to the environment to check if the proposed imaging system could find three of them or not. Figure 11 illustrates the reconstructed image using nine arrays and three tumors in the skin when the breast fat exists. Like the reconstructed image using two tumors, both MWDAS and FDMAS could not detect the three tumors accurately. The TR could discover the three of them, but not their exact locations. However, the improved algorithm detected them very well with only a small alleviation from the actual location.

The next investigation was when the breast had a diameter of 16 cm. Since the system capabilities were checked in other investigations and showed that the improved algorithm could detect it in other evaluations, only the improved algorithm was used for the larger sample. Figure 12 depicts the reconstructed image using the improved algorithm for a breast with a diameter of 16 cm. This shows that the system could detect it perfectly with the most clutter removed. However, little clutter existed around the reconstructed sample at the actual place.

### 4.2. Reconstructed Images Using Measured Data

Figure 13 shows the measurement setup in the presence of both skin and breast. The data were collected using manual switching. Thus, port one of the VNA was connected to the antenna sensor by cable; the second cable was connected to port two of the VNA; and then the cable was connected to the other arrays of the antenna sensor from antenna sensor array 2 to antenna sensor array 9.

The measurement process was done after the fabrication of the skin and the breast using the method presented in the literature. The dielectric properties of the fabricated skin and breast were measured and compared to the presented values in the literature and were shown before starting the imaging reconstruction procedure.

Figure 14 depicts the reconstructed image using the improved algorithm applying the measured data extracted from the measured nine arrays shown in Figure 14. The tumor was located at the center of the fabricated skin and breast fat and the location of X = 35 mm, Y = 0, Z = 15 mm. The tumor was detected at the exact location that was put and inserted in the fabricated skin and breast fat.

A quality of an image can be examined by comparing it with a reference image. The reference image is an image that has no clutter and only contains the tumor image. To check the image quality, the similarity index of the images should be defined to see how much the reconstructed images are different from the reference image (a circular tumor with a diameter of 4 mm at the center of the imaging media). It should be mentioned that the reference image’s mean squared error (MSE) and structural similarity index (SSI) are 0 and 1, respectively. Extra details presented in [43] show how it can be calculated. The SSI factor was calculated for every image and is illustrated in Table 1 when nine array elements existed in all considerations. It was revealed that the SSI was at the highest level when the proposed IMTR algorithm was used.

Table 1 depicts how the proposed IMTR algorithm exceeded the other algorithms such as FDMAS, MWDAS, and conventional TR in terms of the accuracy of the detected location of the tumor. All these algorithms were used to reconstruct the image of the tumor to perform the comparison. First, MWDAS was used since it was an improved version of the conventional DAS algorithm. Then, this was followed by using the FDMAS algorithm. Both these algorithms were upgraded and improved versions of their conventional algorithms. Afterward, the conventional TR algorithm was applied to have a more solid comparison, because the proposed algorithm used the principles of the TR algorithm. When all three algorithms were used for image reconstruction, the proposed algorithm was applied using the same data and received signals. The SSR factor shown in Table 1 illustrates that the images obtained using the proposed algorithm were the most similar to the actual target in terms of shape and location.

Another vital factor to see if an imaging system is working well is to investigate its capability of having a high resolution as they are divided into three types: range, cross-range, and spatial resolution. The related equation to show how they can be calculated is presented in [40,41,42,43]. Applying the equation presented in [44,45,46,47] showed that the proposed system can offer resolution approximate values of 3 mm, 6.9 mm, and 5.3 mm, spatial resolution, cross-range, and range resolution. Table 2 depicts a close comparison between the proposed system and some recent similar studies. It shows that the proposed imaging system had higher gain, wider BW, smaller dimension of antenna, and could detect the smallest tumor compared to the other works presented in Table 2.

## 5. Conclusions

Skin cancer has been identified as one of the major causes of death. This usually happens since the human body is exposed to the sunlight and its UV rays affect the skin negatively. Several techniques have been used to detect skin cancer. However, microwave imaging overtook them in terms of resolution, mobility, accuracy, and being safe. UWB antennas are considered as a vital part of a microwave imaging system. Therefore, the UWB antennas applied for imaging purposes should meet the requirements of medical imaging. These requirements are broad bandwidth, high fidelity factor, and miniaturized dimensions. A UWB antenna was designed with a wide bandwidth of 21 GHz, and a high-fidelity factor of a maximum of 92% at most working bandwidths. The antenna’s performance was investigated in breast skin media and free space when nine array elements of the proposed antenna were located around the breast skin. After showing that the antenna elements worked well in breast skin media, the received signals from the nine array elements were recorded. An improved modified time-reversal algorithm was performed to reconstruct the image of a tumor with a maximum diameter of 3 mm. Various considerations such as a different number of array elements, smaller tumor in skin breast, two and three tumors, and larger breast media were examined to show that the system could detect the tumor perfectly. Furthermore, the system showed that a tumor with a diameter of 3 mm was detected perfectly in all considerations and any locations within the breast skin.

## Figures and Tables

**Figure 1 micromachines-12-00647-f001:**
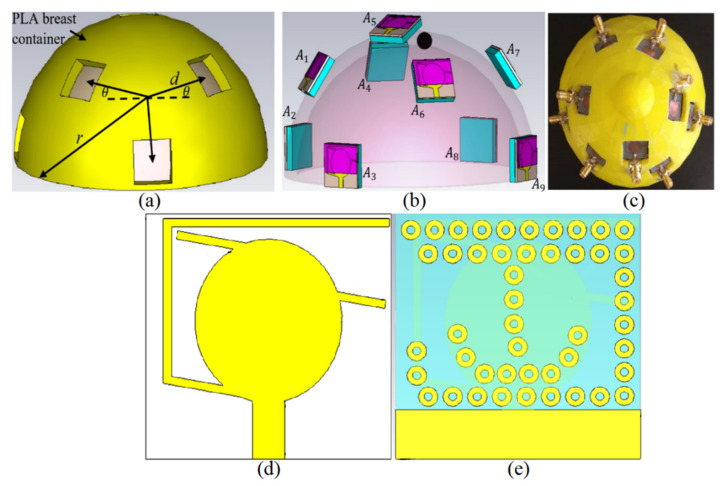
(**a**) modulated UWB antennas around a breast skin media, (**b**) nine simulated prototypes of the antennas around the breast skin, and (**c**) the fabricated phantom, (**d**,**e**) front and back view of the antenna layout [31].

**Figure 2 micromachines-12-00647-f002:**
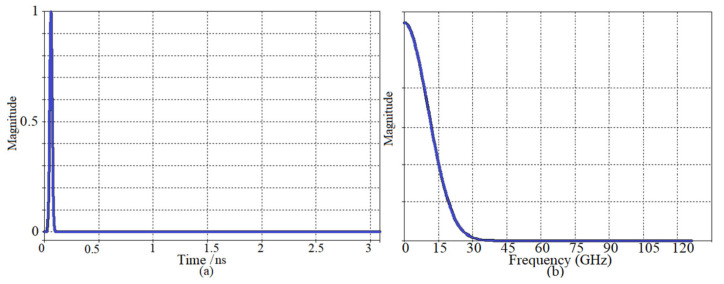
(**a**) The input signal as a Gaussian signal and (**b**) the pulse response of the UWB antenna.

**Figure 3 micromachines-12-00647-f003:**
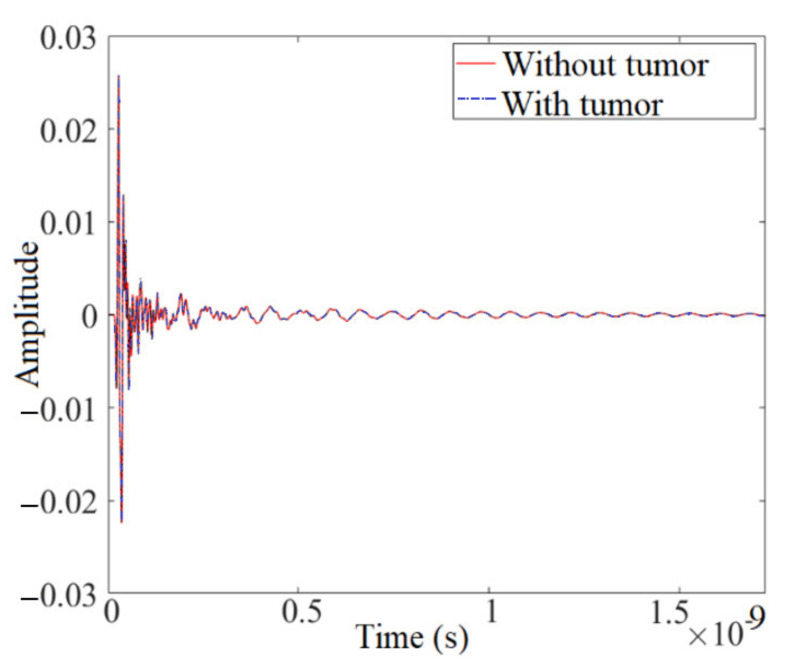
The total received signals.

**Figure 4 micromachines-12-00647-f004:**
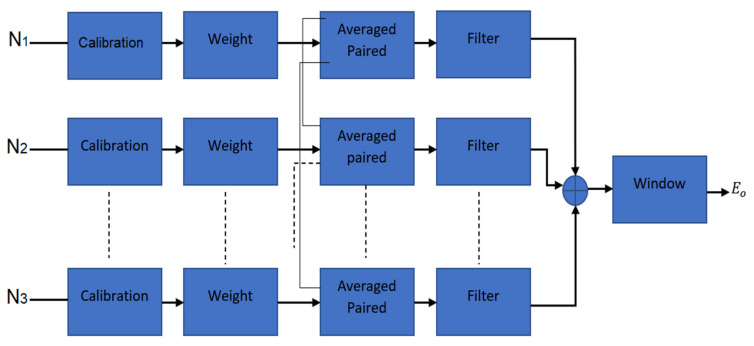
The total received signals.

**Figure 5 micromachines-12-00647-f005:**
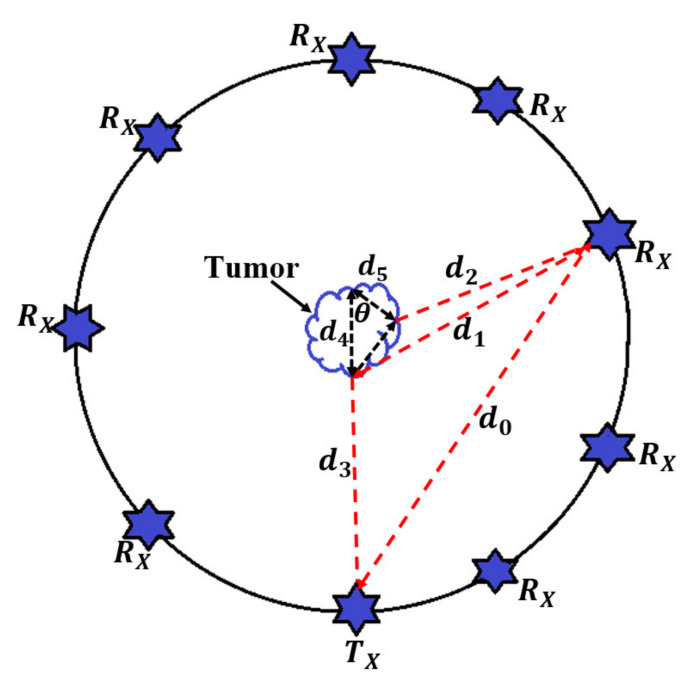
The wave diagram to define the early and late time instants.

**Figure 6 micromachines-12-00647-f006:**
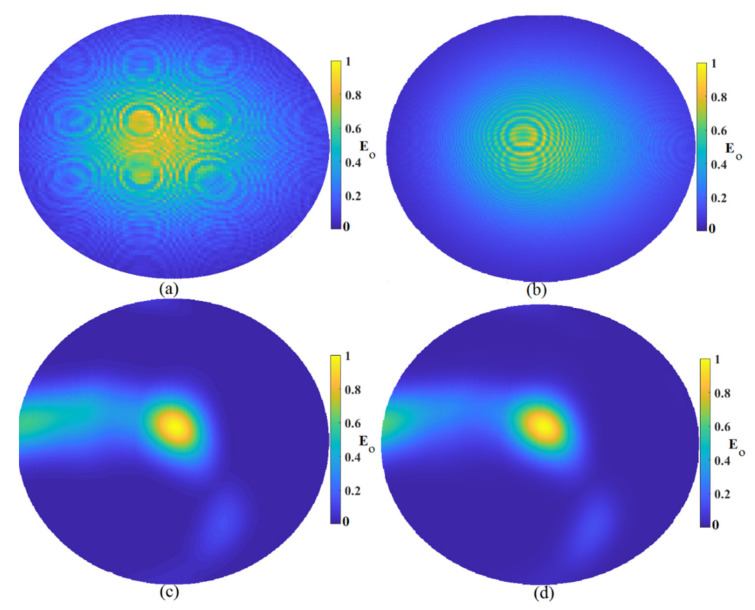
Image reconstruction using three arrays: (**a**) MWDAS, (**b**) FDMAS, (**c**) TR, (**d**) IMTR.

**Figure 7 micromachines-12-00647-f007:**
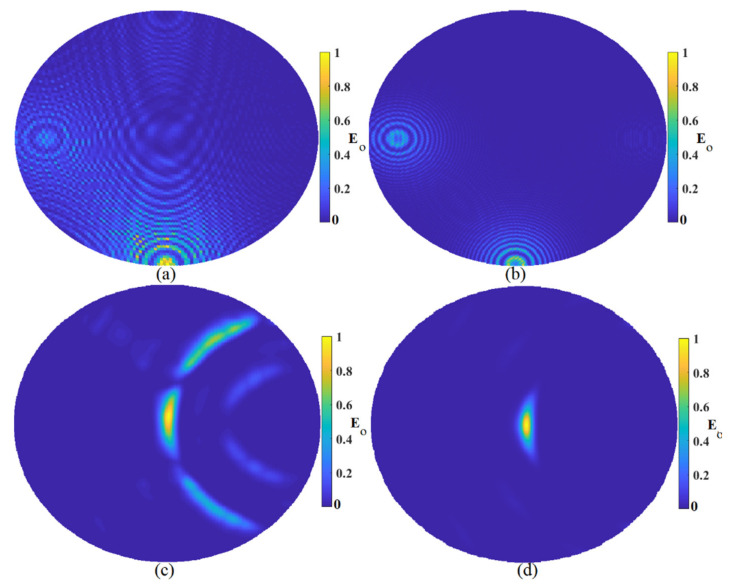
Image reconstruction using six arrays: (**a**) MWDAS, (**b**) FDMAS, (**c**) TR, (**d**) IMTR.

**Figure 8 micromachines-12-00647-f008:**
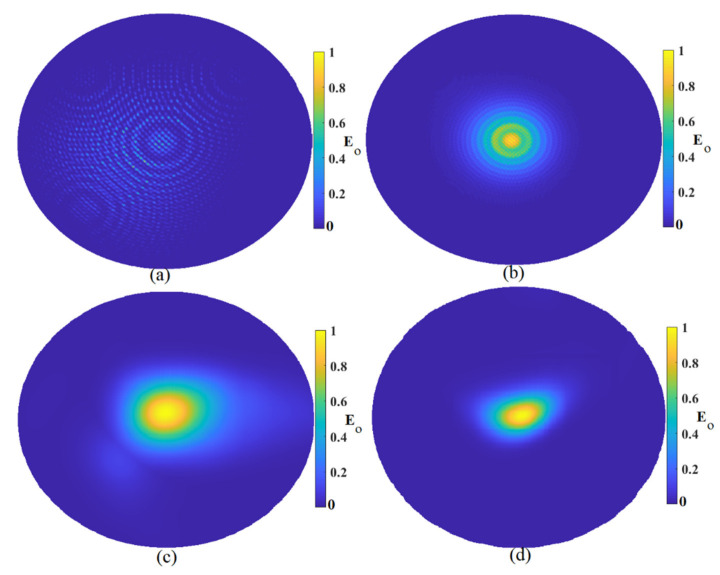
Reconstructed image using nine array elements: (**a**) MWDAS, (**b**) FDMAS, (**c**) TR, (**d**) IMTR).

**Figure 9 micromachines-12-00647-f009:**
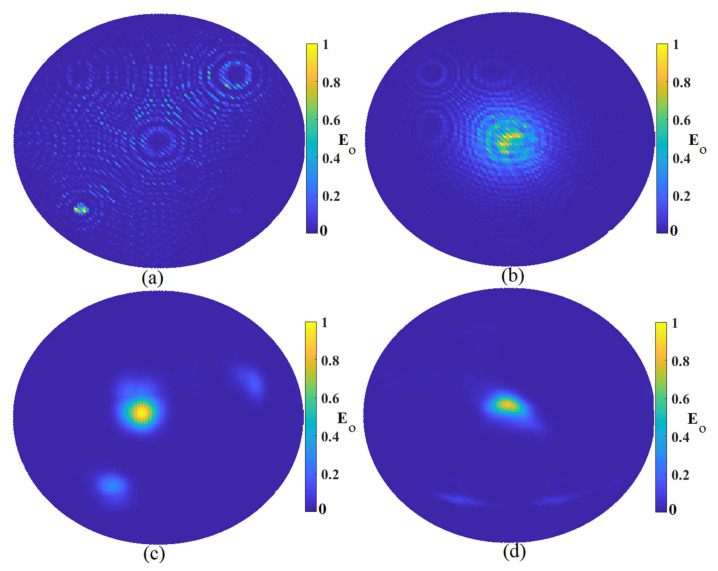
Image reconstruction of a smaller target with a 3 mm diameter: (**a**) MWDAS, (**b**) FDMAS, (**c**) TR, (**d**) IMTR.

**Figure 10 micromachines-12-00647-f010:**
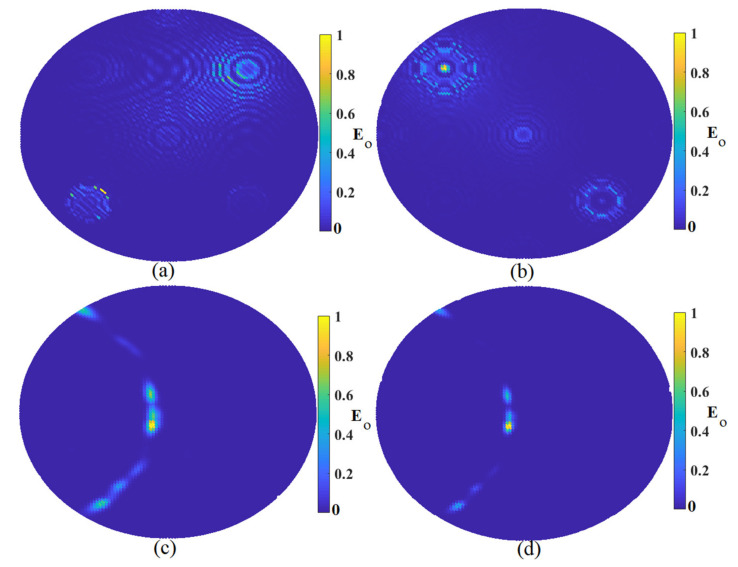
Image reconstruction of two targets: (**a**) MWDAS, (**b**) FDMAS, (**c**) TR, (**d**) IMTR.

**Figure 11 micromachines-12-00647-f011:**
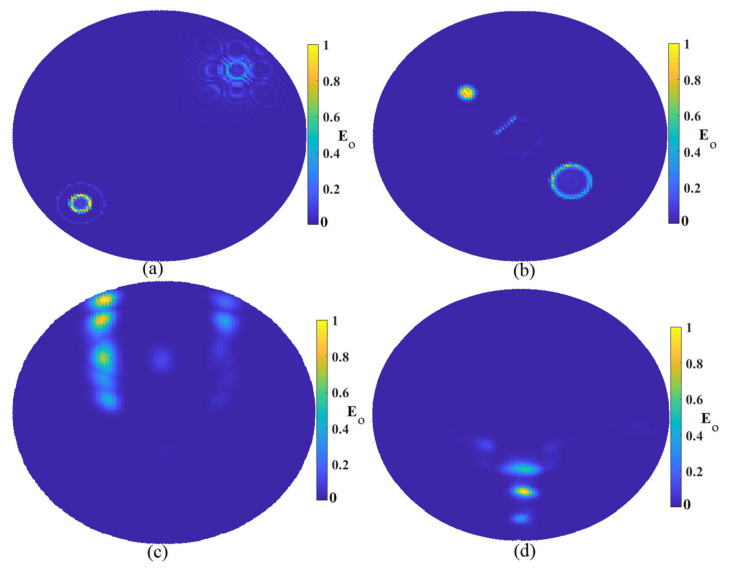
Image reconstruction of three targets: (**a**) MWDAS, (**b**) FDMAS, (**c**) TR, (**d**) IMTR.

**Figure 12 micromachines-12-00647-f012:**
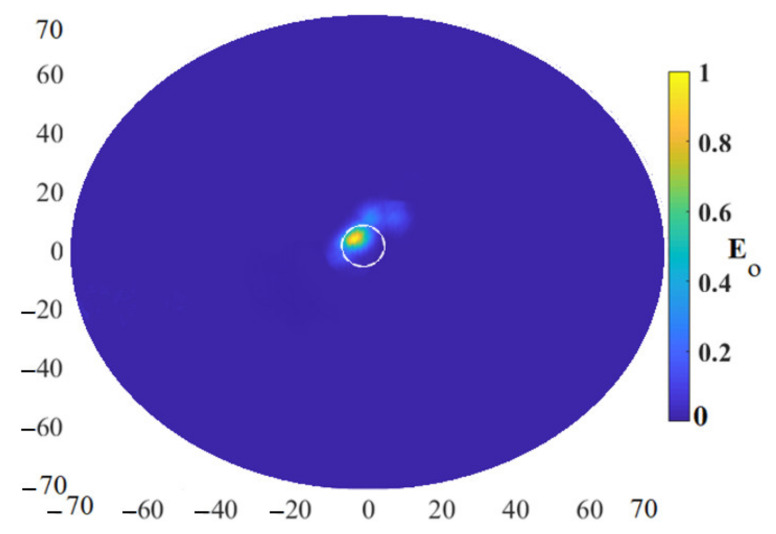
The reconstructed image of a larger breast sample 16 cm in diameter using IMTR.

**Figure 13 micromachines-12-00647-f013:**
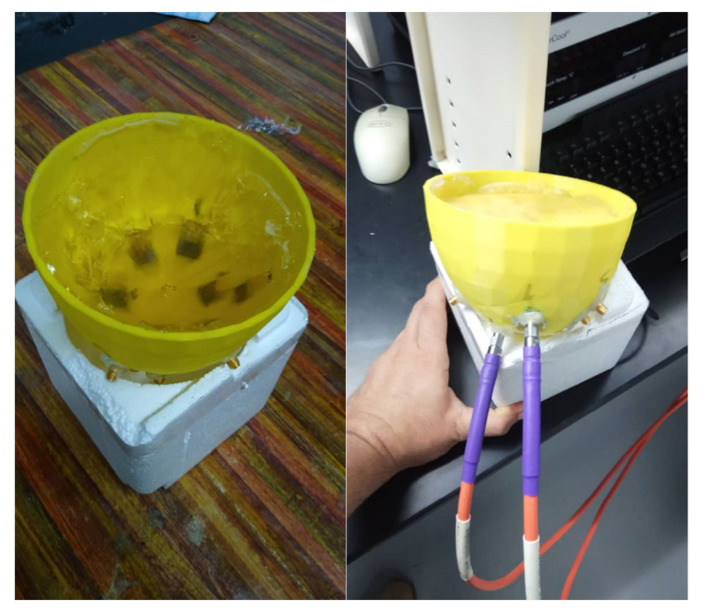
Measurement setup of the breast and the skin.

**Figure 14 micromachines-12-00647-f014:**
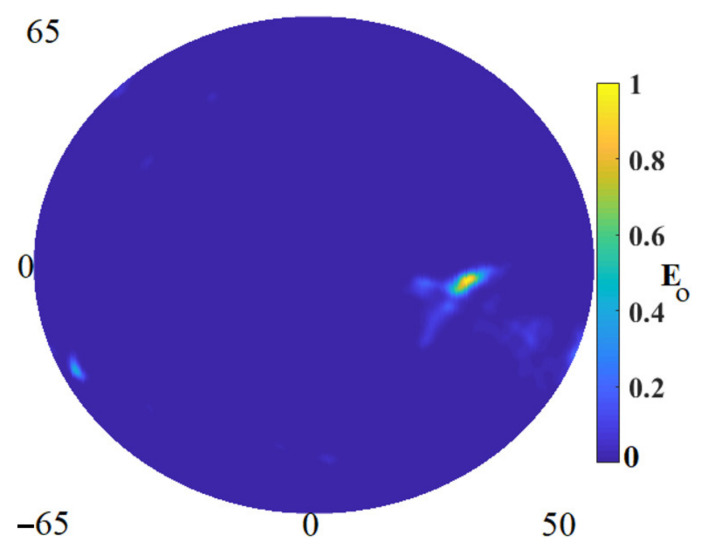
The reconstructed image from the measured data using IMTR.

**Table 1 micromachines-12-00647-t001:** The SSI comparison of the reconstructed images.

Images	SSI	Images	SSI
Figure 6, Figure 7 and Figure 8, proposed	0.9775	Figure 10, TR	0.8567
Figure 6, Figure 7 and Figure 8, TR	0.8756	Figure 10, FDMAS	0.8187
Figure 6, Figure 7 and Figure 8, FDMAS	0.8021	Figure 10, MWDAS	0.7923
Figure 6, Figure 7 and Figure 8, MWDAS	0.7898	Figure 11, pro	0.9843
Figure 9, pro	0.9899	Figure 11, TR	0.9278
Figure 9, TR	0.9679	Figure 11, FDMAS	0.872
Figure 9, FDMAS	0.8943	Figure 11, MWDAS	0.8055
Figure 9, MWDAS	0.74	Figure 12, pro	96.25
Figure 10, pro	0.9687	Figure 14, pro	95.25

**Table 2 micromachines-12-00647-t002:** The Imaging System Comparison with Recent Similar Studies.

Ref	BW (GHz)	Antenna Size (mm^2^)	Tumor Diameter (mm)	Gain (dBi)
[48]	3.74–9.5	28 × 35.5	6	-
[49]	1.1–3.4	70 × 30	10	-
[50]	3.9–8.4	45 × 37	10	6.8
[51]	3–11	33.14 × 15	5	4.74
[52]	3–11	40 × 40	-	6.8
proposed	3.9–30	15 × 15	3–3.5	6.48

## Data Availability

Not applicable.

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
