# Peer review of "Microwave Imaging of Breast Skin Utilizing Elliptical UWB Antenna and Reverse Problems Algorithm"

_micromachines, 2021, doi:10.3390/mi12060647_

Round 1

Reviewer 1 Report

1.In Table 1, the SSI comparison of the reconstructed images should be demonstrated in detail. 2.Please revise and enhance the English writing to increase readability.

Reviewer 2 Report

Authors in this research work have proposed and investigated a miniaturized elliptical ultra-wideband (UWB) antenna and its semi-spherical array arrangement are used for signal transmission and reception that save the scatters from the defected locations in the breast skin. The idea and concept of the are interesting. It has well organized and experimentally validated. However, still there are some comments as listed in below which authors are required to carefully address them prior to final recommendation of consideration.

1) Half of the abstract section is more appropriate for introduction section. Please consider only the research achievements in this part. The proposed algorithm and utilization of the UWB antenna should be more discussed in this part.

2) Since the title and concept of the paper are dealing with UWB antennas, therefore it is required to add a short paragraph in the introduction section providing some information about UWB antennas for various applications along with proper citations. In below there are suitable suggestions.

-"On-Chip Antenna Design Using the Concepts of Metamaterial and SIW Principles Applicable to Terahertz Integrated Circuits Operating over 0.6–0.622 THz" International Journal of Antennas and Propagation, Volume 2020, Article ID 6653095, 9 pages, https://doi.org/10.1155/2020/6653095.

- “Wideband linear microstrip array antenna with high efficiency and low side lobe level” Int J RF Microw Comput Aided Eng. 2020;e22412, doi: 10.1002/mmce.22412

- A Comprehensive Survey of "Metamaterial Transmission-Line Based Antennas: Design, Challenges, and Applications", IEEE Access, vol. 8, pp. 144778-144808, 2020.

-"Enhanced radiation gain and efficiency of a metamaterial inspired wideband microstrip antenna using substrate integrated waveguide technology for sub-6 GHz wireless communication systems", Microw Opt Technol Lett. 2021;1–7. https://doi.org/10.1002/mop.32825.

-“A new class of wideband microstrip falcate patch antennas with reconfigurable capability at circular‐polarization” Microw Opt Technol Lett. 2020; 1– 6, doi: 10.1002/mop.32529.

- A Comprehensive Survey on "Various Decoupling Mechanisms with Focus on Metamaterial and Metasurface Principles Applicable to SAR and MIMO Antenna Systems", IEEE Access, vol. 8, pp. 192965-193004, 2020, doi: 10.1109/ACCESS.2020.3032826.

-"MTM- and SIW-Inspired Bowtie Antenna Loaded with AMC for 5G mm-Wave Applications" International Journal of Antennas and Propagation, Volume 2021, Article ID 6658819, 7 pages https://doi.org/10.1155/2021/6658819.

-“High Efficiency X-Band Series-Fed Microstrip Array Antenna” Progress In Electromagnetics Research C, Vol. 105, 35-45, 2020.

"Metamaterial-Inspired Antenna Array for Application in Microwave Breast Imaging Systems for Tumor Detection", IEEE Access, vol. 8, pp. 174667-174678, 2020, doi: 10.1109/ACCESS.2020.3025672.

3) What is the benefit of section 2? It is very short. Maybe it can be replaced with proper referencing.

4) In Fig.1, the layout of each patch antenna can be separately exhibited to better recognizing its layout.

5) In Fig.2, what about the interaction between the antennas?

6) Before concluding the paper, it would be nice to compare the proposed work with prior arts. Then the results from this comparison can be listed in a table.

7) The performance parameters such as reflection coefficients, transmission coefficients, bandwidth, radiation gain and efficiency can be presented in section 3.1.

8) Section 3.2 is dealing with "modelling of electromagnetic behaviour", how it has done in this work? This section is short.
